# Assessing acceptance of augmented reality in nursing education

**Pelin Uymaz** [ID][1¤a], **Ali Osman Uymaz** [ID][2¤b]*

**1** Department of Nursing, Faculty of Health Sciences, Alanya Alaaddin Keykubat University, Alanya, Antalya, Turkey, **2** Department of Human Resources Management, Faculty of Economics, Administrative and Social Sciences, Alanya Alaaddin Keykubat University, Alanya, Antalya, Turkey

¤a Current address: Alanya Alaaddin Keykubat University, Faculty of Health Sciences, Kestel, Alanya, Antalya, Turkey
¤b Current address: Alanya Alaaddin Keykubat University, Faculty of Economics, Administrative and Social Sciences, Alanya, Antalya, Turkey
* ali.uymaz@alanya.edu.tr

**Data Availability Statement:** The research data is uploaded as a zip file and in the Mendeley Database: UYMAZ, Ali Osman (2021), "Assessing acceptance of augmented reality in nursing education", Mendeley Data, V1, doi: 10.17632/

## Abstract

The Covid-19 pandemic has negatively affected every aspect of human life. In these challenging times nursing students, facing academic and psychological issues, are advised to use augmented reality applications in the field of health sciences for increasing their motivations and academic performances. The main motive of the study was to examine the acceptance status of nursing students in implementing augmented reality technology in their education and training. The study is a quantitative research study, and it uses the causal-comparative screening method. The data used in the study was collected online from 419 nursing students. The hybrid method was preferred. First, the hypotheses based on the linear relationships were defined between the variables which were then tested by the method of structural equation modeling. Second, the method of artificial neural networks was used to determine the non-linear relationships between the variables. The results show that the nursing students have a high intention of using augmented reality technology as a way of self-learning. It was also found that the most emphasized motive behind this intention is the expectation that using augmented reality technology will increase their academic performance. They also think that AR technology has many potential benefits to offer in the future. It was observed that a considerable number of students already use augmented reality technology for its usefulness and with a hedonic motivation. In conclusion, nursing students have a high acceptance of using augmented reality technology during their education and training process. Since we live in a world where e-learning and self-learning education/training have become widespread, it is estimated that students will demand augmented reality applications as a part of holistic education, and as an alternative to traditional textbooks.

## Introduction

Mobile technology devices have evolved rapidly in the last decade. Thanks to the possibility of carrying out many activities simultaneously with different wide-ranging applications, the use

rgck545wp6.1 https://data.mendeley.com/datasets/rgck545wp6/1.

**Funding:** The author(s) received no specific funding for this work.

**Competing interests:** The authors have declared that no competing interests exist.

of mobile technology tools at the individual level has become widespread [1]. This advancement in mobile technology brings changes to the various aspects of life including education [2]. So, we think that adopting mobile technology in nursing education and training is worth exploring. Mobile learning has the potential of significantly improving the nursing students' learning outcomes [3]. It has been discovered that e-learning systems in mobile technologies can help nursing students to study clinical nursing practices [4]. Not only may the incorporation of the innovative technologies assist students or trainees in learning effectively under limited clinical tutoring, but it can also assist healthcare practitioners in conducting training and professional skill reinforcement [5].

Another major factor that has caused changes in every aspect of life is the Covid-19 pandemic. Typical education in classrooms worldwide has been interrupted due to the Covid-19 pandemic after March 2020. The educational and training activities are being carried out online. Both teachers and students try to embrace this change through adopting different tools in online education and training which can be evaluated as crisis management in some ways. The most widely adopted approach is carrying the conventional classroom education to the online platforms. It is expected that by diversifying the education materials, online learning would increase the academic performance of students. The nursing students were advised to download the self-learning Augmented Reality (AR) applications and use them as an auxiliary source while studying clinical nursing practices. AR technology allows the students to study virtual organs, vessels, muscles, and bones. Students can rotate any part of the body 360 degrees and examine them in detail. It offers students the opportunity of gaining experience in clinical practices with virtual intervention to virtual wounds created on the real-like human body [6]. This way, students have the chance to do clinical practices many times over and over [7].

Also, as the students experience the negative consequences of the Covid-19 pandemic[8, 9], it is predicted that AR applications as a self-learning technology [10] can motivate nursing students to study and participate actively in the self-learning [11] and positively influence their learning process and academic performances. As an inevitable outcome of the Covid-19 pandemic, the self-learning approach continues dominating the education system worldwide. For example, in Turkey, the Council of Higher Education has recommended to all universities that 40% of the courses should be carried out online [12].

Although e-learning projects developed based on mobile technology have become widespread throughout the world, they are not yet an alternative to the conventional methods of traditional nursing education. On the other hand, as a self-learning method, AR technology differs from others. Unlike common e-learning tools, it offers more than just a way of transferring knowledge. With AR technology, students do not just passively receive knowledge but also actively practice it.

## Literature review

### Augmented reality

Learners use mobile technology to access learning resources, guidance, and assistance. There have been several successful uses of mobile devices in nursing education. Clinical teachers, for example, use iPads to assist midwives in simulating the clinical practices [13]; health professionals use smartphones and QR codes to foster students' learning [14]. Meanwhile, there has been an increasing focus on mobile technologies in nursing education. For example, Wu [15] used mobile devices to help nursing students improve their knowledge and abilities. Learning of nursing professional knowledge may be significantly boosted with the use of mobile technologies.

AR mobile technology is an important self-learning technology. Merging reality and digital content, it significantly improves the learning process [7], so it has become very important for different fields. AR provides a virtual environment that allows virtual data and images to be attached to real objects [16, 17]. In an AR application, the virtual object providing the information is activated by a mobile device, so the user can use augmented reality objects on the real-life object. The AR technology ensures that the learning process is more effective and efficient than traditional methods and tools [18]. The research findings on AR applications indicate that the active participation of users provides an authentic learning environment and boosts self-learning performance. For example, it was observed that using the AR application makes it easier for students to comprehend the subjects which they had difficulties comprehending with the conventional methods [19]. Moreover, it was found that AR technology provides the skill and experience which come with practice [20–23]. The AR technology does not only transfer knowledge but also helps users to actively make practice [24]. With a highly developed simulation process, the users can drag, transform, enlarge, or reduce images, and change ways of seeing, manipulating, and experimenting [7, 19]. It also provides virtual cases, such as a wound on the human body, so nursing students gain experience in clinical practices [6]. This allows nursing students to repeat the clinical nursing practices as many times as they like without the worry of making a mistake. Azuma and his colleagues [25] define AR technology as an enabling technology. It can also improve learning results by enhancing students' motivation, engagement, and ability, as well as providing them with knowledge persistence [7].

Besides transferring knowledge, AR technology can be considered as a structural tool that provides and improves multiple skills [18], attitudes, and behaviors [20, 26]. For example, surgeons who use the AR application designed for the surgical practices state that besides learning new information they could also have better communication with their colleagues, and they were able to plan better treatment for their patients. The AR application significantly improved the treatment processes and results [20]. The users consider AR applications as an effective way of learning and experiencing the customized and supportive content [24]. When they are compared to the conventional learning tools and practices, the AR applications are superior [27]. The main reason is that they provide an interactive learning environment that makes knowledge, skills, and experience easily accessible with minimum financial cost and time [28]. The users are strongly motivated, and even create an emotional bond with the new technology [29, 30].

## Technology acceptance

Previous studies which were conducted to understand and explain the expectations, intentions, and preferences of new technology users by examining their experiences, attitudes, and behaviors provide crucial information for technology users, developers, and manufacturers [31]. In technology acceptance studies, the intention to use the technology or the factors that affect the current use of that technology are examined [32, 33]. As a result of the research, it has been determined that users' perceptions related to technology, including factors such as perceived usefulness, ease of use, performance expectancy, perceived value, perceived effort, are effective in the acceptance of new technology [33, 34].

AR technology has already attracted attention among scholars and practitioners [35, 36]. Two approaches in the field of education stand out in the use of AR technology. The first approach is task-focused applications that are designed for transferring know-how and skills to specific individuals [21–23]. The second is general applications that aim to entertain while presenting the most basic information. Shortly, it is predicted that besides general AR applications that provide specific skills, AR applications aiming to teach with a holistic approach will also become widespread and equivalent to textbooks in classroom education.

## Research model and hypotheses

At the beginning of this study, the nursing students who were enrolled in an internal medicine nursing course were suggested to download AR applications AsthiAR and SnapLearn on human anatomy from the Google Play Store to their mobile phones and use them as auxiliary resources. None of the students could do their clinical nursing practices internships in 2019 and 2020 because they were postponed and rescheduled to the 2021–2022 academic year due to the COVID 19 pandemic. With the AR technology, the students could examine the human body as a whole and each part in detail. And also, virtual wounds could be created. Students were informed that they could the technology on real human bodies, even on their bodies. This way, they are expected to learn and practice human anatomy, which is important for both the nursing courses and internships. Moreover, the negative effects of the Covid-19 pandemic on the students' education would be as minimized as possible. To examine the acceptance of this new technology by the students, the following research questions were determined: "Do nursing students intend to use AR technology in the future to learn a subject?", "Which factors affect the nursing students' use of AR technology?".

So, this study examines the nursing students' acceptance and use of AR technology as an auxiliary resource in their education. The conceptual framework of the study is based on the Unified Theory of Acceptance and the Use of Technology (UTAUT) model which was developed by Venkatesh and his colleagues [33]. UTAUT is a model for examining both behavioral intention (BI) and user behavior (UB) towards information technology [37]. The UTAUT model, including both BI and UB, was preferred since the students gained experience as AR users.

As seen in Fig 1, the research model includes performance expectancy (PE), effort expectancy (EE), facilitating conditions (FC), social influence (SI), perceived value (PV), and anxiety (A) as variables that affect the behavioral intention (BI). The factors which affect the use behavior (UB) include behavioral intention (BI), usefulness (U), and hedonic motivation (HM). The core relationships theorized by UTAUT and HM have a mediation effect on the relationship between U and UB.

*Performance Expectancy* (PE) is defined as the user's expectancy of reaching the desired performance [33]. In previous studies on the acceptance of technology, some found a significant and positive relationship between BI and PE [38, 39], while others could not determine any statistically significant relationship between them [40]. In the context of this study, PE is

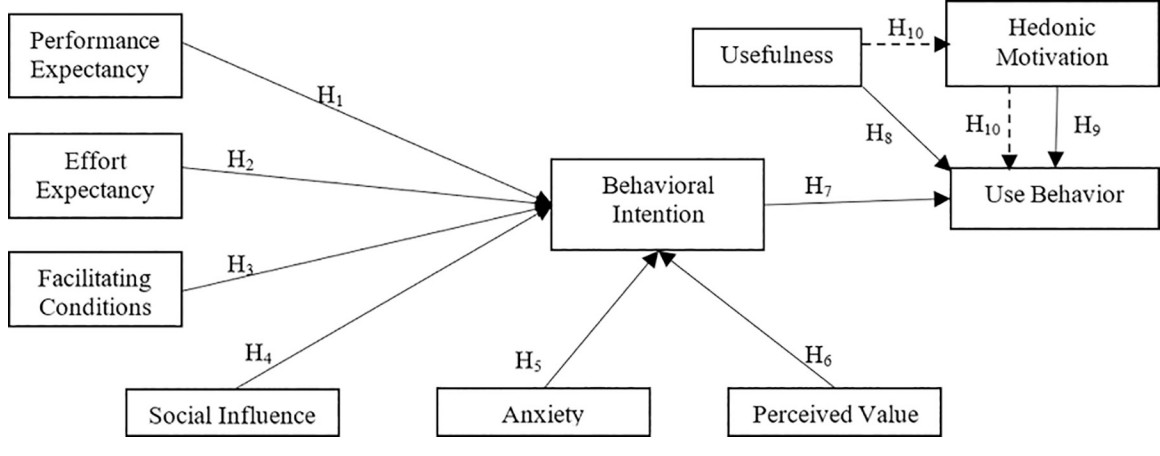

**Fig 1. Research model.**

considered as the expectation of increasing the academic performance and clinical nursing practices created by the use of AR technology in nursing students.

**H₁** PE has a positive and significant influence on the BI of using AR.

*Effort Expectancy* (EE) is the state of effort and ease of usage that the user has to manifest in using the technology [33]. Some studies found a positive relationship between PE and BI [41], while no significant relationship was found in others [39]. In the context of this study, EE is defined as the effort that nursing students have to show in using AR technology.

**H₂** EE has a positive and significant influence on BI of using AR.

*Facilitating Conditions* (FC) is the degree of possessing the resources and support necessary for users to be able to reach the technology [33, 37]. The user's ability to have sufficient resources, necessary knowledge, and help when needed facilitates the adoption and usage of the technology [41, 42]. In some studies, no significant relationship was found between FC and BI [39, 40]. In the context of this study, FC is defined as the effect of the support nursing students can acquire and the necessary information they have while using AR technology.

**H₃** FC has a positive and significant influence on BI of use AR.

*Social Influence* (SI) is about the effect of others on the user's usage and intention to use technology [33]. Some studies found a significant relationship between SI and BI in terms of technology acceptance [43, 44], while no significant relationship between them was found in other studies [38, 39]. In the context of this study, SI is defined as the effect of their social environment on nursing students' intention to use AR technology.

**H₄** SI has a positive and significant influence on BI of use AR.

*Anxiety* (A) has been discussed as the stress caused by using new technology on users in the technology acceptance literature [45]. The concept of anxiety has been taken into consideration within a wide range such as technology anxiety [46, 47], mobile anxiety [48], and computer anxiety [49]. The common ground of these studies is that individuals encounter a new technology and experience stress as a result of using it. In the context of this study, A is defined as the stress that nursing students have on using AR technology as new technology.

**H₅** A has a negative and significant influence on BI of use AR

*The Perceived Value* (PV) is that the benefits of the technology outweigh the costs that the user has to endure to use the technology [37]. The user's perception of technology as "good value for money" is an indicator of having the intention to bear the costs of using the technology [50]. Some studies found a significant relationship between PV and BI [40, 51], whereas no significant relationship was found in other studies [39]. In the context of this study, PV is defined as the nursing students' assessment of whether the benefits of AR technology will be worth the money they will pay.

**H₆** PV has a positive and significant influence on the BI of using AR.

*Behavioral Intention* (BI) is the user's attitude towards the technology. It is the intention that shows whether s/he perceives the related technology positively or negatively and whether s/he wants to use it or not. Even though the attitude is not the behavior itself, it is an important factor to predict the behavior [52]. The studies have found that the attitude of BI has an important regulatory effect on UB [37, 41, 53]. In this study, BI is the intention of nursing students to use AR technology.

**H₇** BI has a positive and significant influence on the UB of AR.

*Usefulness* (U) is that the technology used is compatible with the related ecosystem meets the needs of the user and increases his/her performance. According to Davis [32], usefulness is a critical factor in the acceptance and spread of a system. Similarly, the results of various studies confirm the premise that the perception of usefulness is an important factor in the acceptance and the use of technology [54, 55]. It has been observed that the flexibility and interactive environment provided by the AR application in personal mobile devices encourage

and strengthen the user behavior [24]. In the context of this study, if the students are experienced users of AR, U is defined as suitable for the health education ecosystem, meeting the needs of nursing students and improving their performance.

**H$_8$** U has a positive and significant influence on the UB of AR.

*Hedonic Motivation* (HM) is the feeling of fun and pleasure of using a technology [37]. The emotions resulting from personal experiences are considered effective in adapting to the technology [56]. It was also found that positive emotions reinforce technology acceptance and motivation to use it [29, 57]. HM is defined as one of the main factors affecting behavioral intention [58].

At the same time, the HM of users is determined as an important factor in UB [24]. Considering the AR technology user experiences of the nursing students, evaluating the direct influence of HM on UB was included in the research model.

**H$_9$** HM has a positive and significant influence on the UB of AR.

Users enjoy the usefulness (U) feature of the AR technology and prefer to use it [24]. So, the U feature is a motivator. It is predicted that U has an indirect influence on UB through HM, that is, HM plays a mediator role in the relationship between U and UB.

**H$_{10}$** HM has a mediation effect on the relationship between U and UB.

## Research method

The study was designed according to the causal comparison screening method which is preferred when there is a need to determine the cause-effect relationships of individuals' attitudes, behaviors, ideas, and beliefs [59]. The hybrid method was adopted. First, the hypotheses were tested with Structural Equation Modeling (SEM). SEM is an analysis method used for analyzing the linear relationships between the variables defined in the research model. One disadvantage of using SEM is the possibility of ignoring the unforeseen relationships in the research model because it analyzes through defined relationships [60]. Therefore, as a second step, the nonlinear relationships between the variables were tested with Artificial Neural Network (ANN) to compensate for the disadvantage of SEM. ANN is an analysis method that is inspired by the human nervous system. It enables learning by example from the representative data describing a physical occurrence or a decision process. ANN is distinguished by its ability to build empirical relationships between independent and dependent variables, as well as extract delicate and complicated information from representative data sets. ANNs must be trained in many cases. By training, the relationships between independent and dependent variables can be constructed without making any assumptions regarding a research model [61]. Using ANN also gives a chance to verify the results of SEM. Since it is difficult to develop a hypothesis according to the ANN method, it is recommended to use the hybrid model instead of solely the ANN method [39, 62].

## Sample and data

This study has been carried out following The Code of Ethics of the World Medical Association (Declaration of Helsinki) for experiments involving humans. Ethical approval was obtained from the Human Sciences Scientific Research and Publication Ethics Committee of Alanya Alaaddin Keykubat University, Turkey (reference number E-70561447-050.01.04–35758).

The target population of this study is nursing students. The data was collected between November 2020 and January 2021 online from the nursing students who have enrolled in an internal medicine nursing course. None of the participants could do their clinical nursing practices internships in 2019 and 2020 because they were postponed and rescheduled to the

2021–2022 academic year due to the COVID 19 pandemic. 446 students voluntarily participated in the study. They were informed that the data will be used for only scientific purposes and their personal information will be kept confidential. Of the 446 administered surveys, 27 were excluded due to being improperly completed. The sample includes 310 female and 109 male students, 63 grad and 356 undergrad students, and the average age is 21.

## Scales used in data collection

The study uses the UTAUT model developed by Venkatesh and his colleagues [37], the scales used in the data collection were adapted for the expressions of PE, EE, SI, FC, HM, BI, PV, while the factor of UB was adapted for the expressions of habit factor. The Expressions of the U construct were adapted from the TAM model developed by Venkatesh and his colleagues [33]. In this study, the UTAT model was adapted for analyzing the acceptance and use of information technology [37], intelligent learning system [63], and intelligent healthcare systems [64]. All indicators were measured by a five-point Likert type scale ranging from 1 (refers to "strongly disagree") to 5 (refers to "strongly agree").

In the first stage of the process, SEM from multidimensional and linear analysis methods was applied. The tests were carried out with SmartPLS 3.2.9 statistical package program. In the second stage, SPSS 24 statistical package program was used for the ANN test.

## Analysis

### Model validation

In the preliminary analysis of the research model, as suggested by Hair and his colleagues [65], the convergent validity, internal consistency, and discriminant validity analyses were respectively performed for reflective and formative measurement models.

For indicator reliability, the indicators with loading values less than 0.40 were excluded from the analysis while the indicators with loading values above 0.70 were kept. The effects of the indicators with loading values between 0.40 and 0.70 on the values of AVE, Cronbach's Alpha, and CR were examined, when they were excluded from the analyses. As seen in Fig 2, some indicators with outer loading values less than 0.70 that meet the norm values were involved in the analyses too.

It is suggested that the values of Cronbach's alpha and composite reliability (CR) should be higher than 0.70 for internal consistency. The values of Cronbach's Alpha, CR, and AVE which are the indicators of internal consistency of the constructs met the requirements as seen in Table 1.

After analyzing the internal consistency and convergent validity, Fornell-Larker [66] and Heterotrait-Monotrait [67] (HTMT) tests were conducted for analyzing the discriminant validity.

According to the Fornell-Lacker method, for discriminant validity between the constructs, the diagonal values of each construct should be greater than the values of other elements in the same column. The Fornell-Lacker test results given in Table 2 indicate that each construct confirms discriminant validity and that the constructs have distinctive validity in the scope of the research model.

Researchers such as Henseler and his colleagues [67] argued that the Fornell-Lacker method is not sensitive in terms of discriminant validity, which is why HTMT should be used as a second test. As seen in Table 2 of the analysis results, it is observed that the model has a discriminant validity since the values of HTMT between the constructs are equal to or less than 0.90.

Before conducting the research model path analysis, the values of Variance Inflation Factor (VIF) were evaluated for collinearity analysis between the indicators. If the values of VIF equal

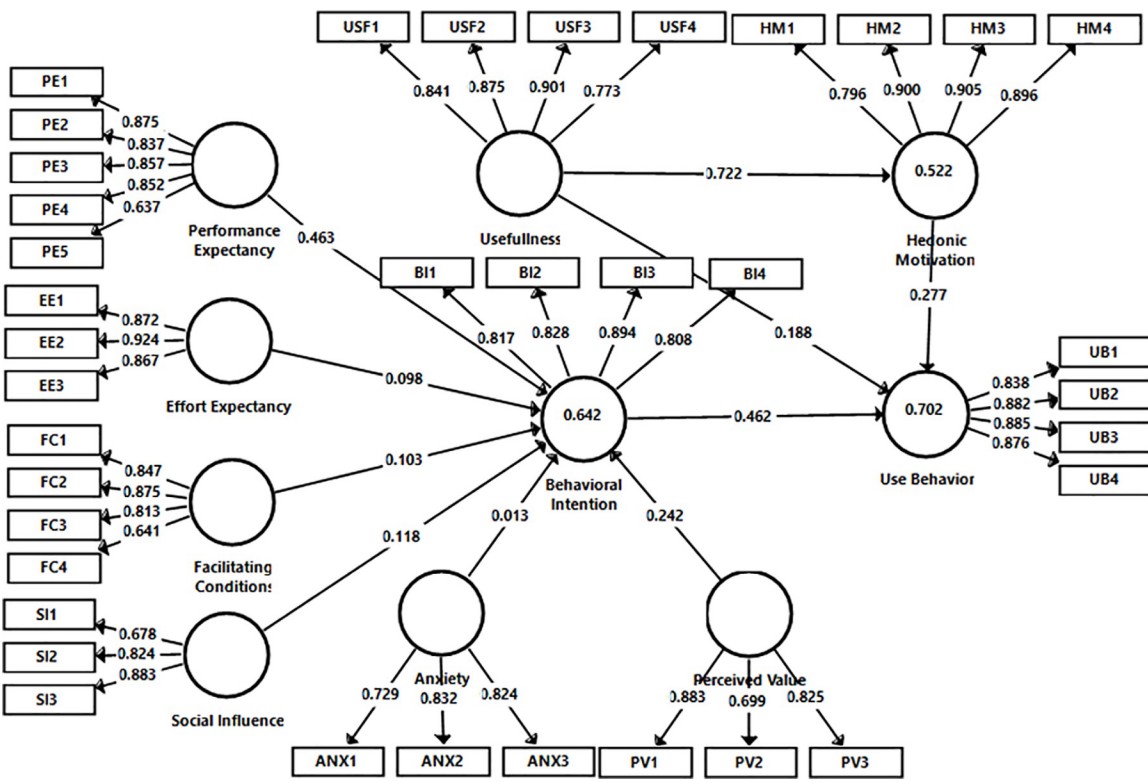

**Fig 2. The research model path analysis.**

to or above 5 indicate that the model may be contaminated by common method bias [65]. As seen in Table 1, the VIF values of the indicators are found to change in a range from 1.276 to 3.387 and the collinearity between indicators used in the research model analysis was considered non-problematic.

## The research model evaluation

In the research model analysis, according to the sequence suggested by Hair and his colleagues [65], the size and significance of path coefficients, coefficients of determination ($R^2$), and predictive relevance ($Q^2$) were carried out.

For the research model path and inter-factor significance analysis, the bootstrapping method was applied with regenerated 5.000 samples. For the 0.05 significance level, the value of p should be less than 0.05, and the t-test value should be above 1.96 [65].

As can be seen in Table 3 all hypothesized path relationships were supported except for $H_6$.

One of the most important coefficients used in the evaluation of the research model is $R^2$, the coefficient determination. The value of $R^2$ shows the total effect of exogenous latent constructs on endogenous latent constructs. $R^2$ values of 0.67, 0.33, and 0.19 were defined as respectively high, medium, and low-level thresholds [68]. As seen in Table 3, and Fig 2 the explanation level of exogenous constructs with $R^2$ value 0.702 of UB construct is high, whereas its explanation level with $R^2$ value 0.522 of HM and $R^2$ value 0.642 of BI are medium.

In addition to $R^2$ analysis, Hair and his colleagues [65] have proposed an $f^2$ test to determine the effect of exogenous constructs on endogenous constructs by testing the effect of endogenous construct on $R^2$, when the exogenous construct contributing to the $R^2$ value is excluded from the research model. The formula used in the analysis is $f^2 = R^2$ Included$-R^2$

**Table 1. Averages, standard deviations, composite reliability, internal consistency, multicollinearity analysis results.**

| | Mean | SD | Loadings | VIF | Cronbach's Alpha | CR | AVE |
|---|---|---|---|---|---|---|---|
| **Behavioral Intention** | **3.873** | **0.872** | | | **0.858** | **0.904** | **0.701** |
| I intend to use the AR in the future | 3.685 | 0.973 | 0.817 | 1.851 | | | |
| I predict I would use the AR in the future | 4.260 | 0.917 | 0.828 | 1.909 | | | |
| I plan to use the AR in the future | 3.790 | 1.022 | 0.894 | 2.628 | | | |
| I will try to use AR in my daily life | 3.757 | 0.969 | 0.808 | 1.923 | | | |
| **Effort Expectancy** | **3.789** | **0.958** | | | **0.865** | **0.918** | **0.789** |
| My interaction with the AR would be clear and understandable | 3.752 | 0.982 | 0.872 | 2.230 | | | |
| It would be easy for me to become skillful at using the AR | 3.749 | 0.988 | 0.924 | 2.924 | | | |
| I would find the AR easy to use | 3.866 | 0.904 | 0.867 | 2.085 | | | |
| **Facilitating Conditions** | **3.509** | **1.127** | | | **0.810** | **0.875** | **0.639** |
| I have the resources necessary to use the AR | 3.494 | 1.189 | 0.847 | 2.082 | | | |
| I know necessary to use the AR | 3.339 | 1.218 | 0.879 | 2.423 | | | |
| A specific person (or group) is available for assistance with AR difficulties | 3.804 | 1.032 | 0.813 | 1.614 | | | |
| The AR is not compatible with other systems I use | 3.399 | 1.069 | 0.641 | 1.342 | | | |
| **Hedonic Motivation** | **4.084** | **0.948** | | | **0.898** | **0.929** | **0.766** |
| Using the AR is fun | 4.158 | 0.934 | 0.796 | 1.901 | | | |
| Using the AR is enjoyable | 4.186 | 0.913 | 0.900 | 2.873 | | | |
| Using the AR is very entertaining | 3.983 | 0.968 | 0.905 | 3.373 | | | |
| Using the AR is very attractive | 4.007 | 0.976 | 0.896 | 3.387 | | | |
| **Perceived Value** | **3.547** | **1.069** | | | **0.730** | **0.846** | **0.649** |
| The AR applications are reasonably priced | 3.699 | 0.999 | 0.883 | 1.721 | | | |
| The AR is a good value for the money | 3.148 | 1.136 | 0.699 | 1.323 | | | |
| At the current price. the AR provides a good value | 3.795 | 1.073 | 0.825 | 1.521 | | | |
| **Performance Expectancy** | **4.229** | **0.872** | | | **0.871** | **0.908** | **0.667** |
| I would find AR useful in nursing education. | 4.358 | 0.812 | 0.875 | 2.863 | | | |
| I would find AR useful in my courses. | 4.434 | 0.807 | 0.837 | 2.732 | | | |
| Using AR enables me to understand subjects more quickly. | 4.305 | 0.864 | 0.857 | 2.466 | | | |
| Using the AR increases my skills. | 4.451 | 0.846 | 0.852 | 2.294 | | | |
| If I use the AR. I will increase my chances of getting better grades. | 3.599 | 1.032 | 0.637 | 1.359 | | | |
| **Social Influence** | **3.974** | **0.976** | | | **0.719** | **0.840** | **0.640** |
| People who influence my behavior think that I should use the AR | 3.556 | 1.107 | 0.678 | 1.307 | | | |
| People who are important to me think that I should use the AR | 4.317 | 0.861 | 0.824 | 1.490 | | | |
| People whose opinions that I value prefer that I use the AR | 4.048 | 0.961 | 0.883 | 1.724 | | | |
| **Anxiety** | **3.791** | **0.975** | | | **0.711** | **0.838** | **0.634** |
| I feel apprehensive about using the AR | 3.737 | 0.974 | 0.729 | 1.276 | | | |
| It scares me to think that I could lose a lot of information using the AR by hitting the wrong place | 3.938 | 0.938 | 0.832 | 1.469 | | | |
| The AR is somewhat intimidating to me | 3.697 | 1.014 | 0.824 | 1.539 | | | |
| **Use Behavior** | **4.110** | **0.907** | | | **0.893** | **0.926** | **0.758** |
| Using AR applications is like a habit for me. | 4.274 | 0.854 | 0.838 | 2.078 | | | |
| AR applications are a natural learning environment for me. | 4.064 | 0.944 | 0.882 | 2.696 | | | |
| Whenever I want. using AR applications make it easier for me | 3.971 | 0.939 | 0.885 | 2.758 | | | |
| I prefer to use AR applications while studying. | 4.129 | 0.889 | 0.876 | 2.463 | | | |
| **Usefulness** | **4.133** | **0.974** | | | **0.869** | **0.911** | **0.721** |
| I find the AR products useful in health education | 4.026 | 1.021 | 0.841 | 2.118 | | | |
| Using the AR product would improve my daily studies performance | 4.010 | 0.975 | 0.875 | 2.966 | | | |
| Using the AR product helped to improve my professional knowledge. | 4.103 | 0.946 | 0.901 | 3.269 | | | |
| Using the AR product would enhance effectiveness in the health education courses. | 4.394 | 0.823 | 0.773 | 1.556 | | | |

**Table 2. Fornell-Lacker and HTMT discriminant validity analyses.**

| | Fornell-Lacker | | | | | | | | | | HTMT | | | | | | | | |
|---|---|---|---|---|---|---|---|---|---|---|---|---|---|---|---|---|---|---|---|
| | BI | EE | FFC | HM | PV | PE | SI | A | UB | U | BI | EE | FFC | HM | PV | PE | SI | A | UB | U |
| BI | 0,84 | | | | | | | | | | | | | | | | | | | |
| EE | 0,49 | 0,89 | | | | | | | | | 0,57 | | | | | | | | | |
| FC | 0,38 | 0,30 | 0,80 | | | | | | | | 0,45 | 0,36 | | | | | | | | |
| HM | 0,70 | 0,39 | 0,42 | 0,88 | | | | | | | 0,79 | 0,45 | 0,49 | | | | | | | |
| PV | 0,63 | 0,36 | 0,38 | 0,68 | 0,81 | | | | | | 0,77 | 0,44 | 0,48 | 0,82 | | | | | | |
| PE | 0,72 | 0,44 | 0,25 | 0,59 | 0,52 | 0,82 | | | | | 0,83 | 0,51 | 0,29 | 0,67 | 0,64 | | | | | |
| SI | 0,55 | 0,53 | 0,33 | 0,62 | 0,51 | 0,47 | 0,80 | | | | 0,68 | 0,68 | 0,44 | 0,76 | 0,68 | 0,56 | | | | |
| A | 0,49 | 0,29 | 0,45 | 0,57 | 0,61 | 0,43 | 0,46 | 0,80 | | | 0,62 | 0,37 | 0,59 | 0,71 | 0,84 | 0,55 | 0,63 | | | |
| UB | 0,79 | 0,52 | 0,34 | 0,74 | 0,58 | 0,69 | 0,57 | 0,49 | 0,87 | | 0,89 | 0,59 | 0,39 | 0,82 | 0,69 | 0,78 | 0,70 | 0,61 | | |
| U | 0,70 | 0,37 | 0,27 | 0,72 | 0,67 | 0,74 | 0,54 | 0,58 | 0,71 | 0,85 | 0,80 | 0,42 | 0,31 | 0,81 | 0,82 | 0,84 | 0,65 | 0,73 | 0,81 | |

Note. PE: performance expectancy; EE: effort expectancy; FC: facilitating conditions; SI: social influence; PV: perceived value; A: anxiety; BI: behavioral intention; U: usefulness; HM: hedonic motivation; UB: use behavior.

$_{Excluded}$ / 1 −R$^2$ $_{Included}$. In the evaluation of the test results, 0.02, 0.15, and 0.35 threshold values were defined as a small, medium, and large effect [65, 69]. As seen in Table 3, the effect of PE (with the value of 0.37 f$^2$) on BI was large and the effects of other constructs on BI were small. It is also shown that the effect of the BI construct (with the value of 0.52 f$^2$) on the UB factor was large and the effects of other constructs on UB were small.

The Stone-Geisser's Q$^2$ values were calculated by running the Blindfolding test to evaluate the prediction power of the research model. The value of Q$^2$ represents the out-of-sample

**Table 3. The research model evaluation results.**

| | B | R$^2$ | f$^2$ | Q$^2$ | t- Statistics | P | Hypothesis |
|---|---|---|---|---|---|---|---|
| BI | | | | 0.420 | | | |
| HM | | | | 0.375 | | | |
| U | | | | 0.499 | | | |
| Path Coefficients | | | | | | | |
| PE -> BI | 0.463 | 0.642 | 0.372 | | 10.679 | 0.000*** | H$_1$ supported |
| EE -> BI | 0.098 | | 0,014 | | 2.523 | 0.012** | H$_2$ supported |
| FC -> BI | 0.103 | | 0.019 | | 2.852 | 0.004** | H$_3$ supported |
| SI -> BI | 0.118 | | 0.019 | | 3.048 | 0.002** | H$_4$ supported |
| A -> BI | 0.013 | | -0.003 | | 0.300 | 0.764 | H$_5$ not supported |
| PV -> BI | 0.242 | | 0.080 | | 5.390 | 0.000*** | H$_6$ supported |
| BI -> UB | 0.462 | 0.702 | 0.519 | | 8.876 | 0.000*** | H$_7$ supported |
| U -> UB | 0.188 | | 0.042 | | 7.656 | 0.000*** | H$_8$ supported |
| HM -> UB | 0.277 | | 0.088 | | 4.909 | 0.000*** | H$_9$ supported |
| Specific Indirect Effect (Mediator Effect) | | | | | | | |
| U -> HM -> UB | 0.200 | | | | 5.045 | 0.000*** | H$_{10}$ supported |

Note. PE: performance expectancy; EE: effort expectancy; FC: facilitating conditions; SI: social influence; PV: perceived value; A: anxiety; BI: behavioral intention; U: usefulness; HM: hedonic motivation; UB: use behavior.

* 0.10

** 0.05

*** 0.000 at significance level

predictive power of the research model. In the blindfolding method, the value of $Q^2$ is calculated by giving a specific value of omission distance. As a technique for reusing the sample in the analysis, by eliminating each seventh data in the endogenous construct (in this study, omission distance is 7), the predictive power for endogenous construct is analyzed with the remaining data [65, 68, 70]. $Q^2$ value greater than 0 indicates that the research model has predictive relevance for a certain endogenous construct; a $Q^2$ value equal to or less than 0 indicates a lack of predictive relevance for endogenous construct [71]. In the analysis of $Q^2$ values, the values less than 0.15 have a small effect, the values between 0.15 and 0.35 have a medium effect, and the values above 0.35 have a large effect as threshold values. As seen in Table 3 all values of $Q^2$ are above 0.35. That is, the research model has a large predictive power in terms of endogenous constructs.

## The artificial neural network analysis

The artificial neural network (ANN) analysis is defined as information resulting from a learning process like the working system of the human brain [72]. The ANN analysis uses feed-forward, back-propagation, multilayer perception methods in understanding, explaining, and predicting a dependent variable with independent variables [39, 62, 72]. The information derived from the results of the learning process is stored in the synaptic weights [55, 60]. It was observed that the ANN method showed better performance, according to the research methods such as logistic, linear regression, and discriminant analysis [62, 72–74]. It is recommended that by using 70% of the data in the ANN model test as model training and 30% is as model testing, 10 fold cross-validation process will be sufficient for generating a result with a minimum level error [39, 62]. In addition to this, it is also suggested that statistically significant independent constructs and dependent constructs should be added to the test in the linear model SEM analysis [60, 75].

ANN works as a feed-forward and backpropagation, multi-layer perceptron consisting of three layers as input, hidden, and output [76]. It is the modeling of the biological neurons in the brain and the synaptic link established between each other by a computer program. In the learning process, neurons form networks by connecting to each other in various ways. These networks have the ability to learn, memorize and reveal the relationship between variables [62, 77]. In Fig 1. Research model, two sub-models were generated for the ANN test in the context of the research model. As seen in Fig 3, Model A has five neurons in the input layer. Each neuron is represented by independent constructs which are performance expectancy, effort expectancy, facilitating conditions, social influence, and perceived value. Behavioral intention, the dependent construct has been represented by one neuron in the output layer. Behavioral intention, as seen in Fig 4, Model B has three neurons in the input layer. Each neuron is represented by independent constructs which are behavioral intention, usefulness, and hedonic motivation. Use behavior, a dependent construct, has been represented by one neuron in the output layer. As seen in Figs 3 and 4, Bias, H(1:1), H(1:2)... are the hidden layer of ANN.

As seen in Tables 4 and 5, ANN models were trained ten times. By training, as seen in Fig 3 Model A and Fig 4 Model B, the relationships between independent and dependent constructs can be constructed without making any assumptions regarding a research model [61]. The root-mean-square error (RMSE) values are used to evaluate the ANN model analysis results. The results show that the RMSE values for both models are quite small, as seen in Table 4. The average RMSE value for Model A is 0.094 and the average RMSE value for Model B is 0.096. These results are considered quite accurate and satisfactory [60, 73].

After having significant RMSE values, the significance analysis of the independent constructs in the ANN models was performed [60, 62]. The significance analyses are made through the loadings and the normalized importance of the independent constructs.

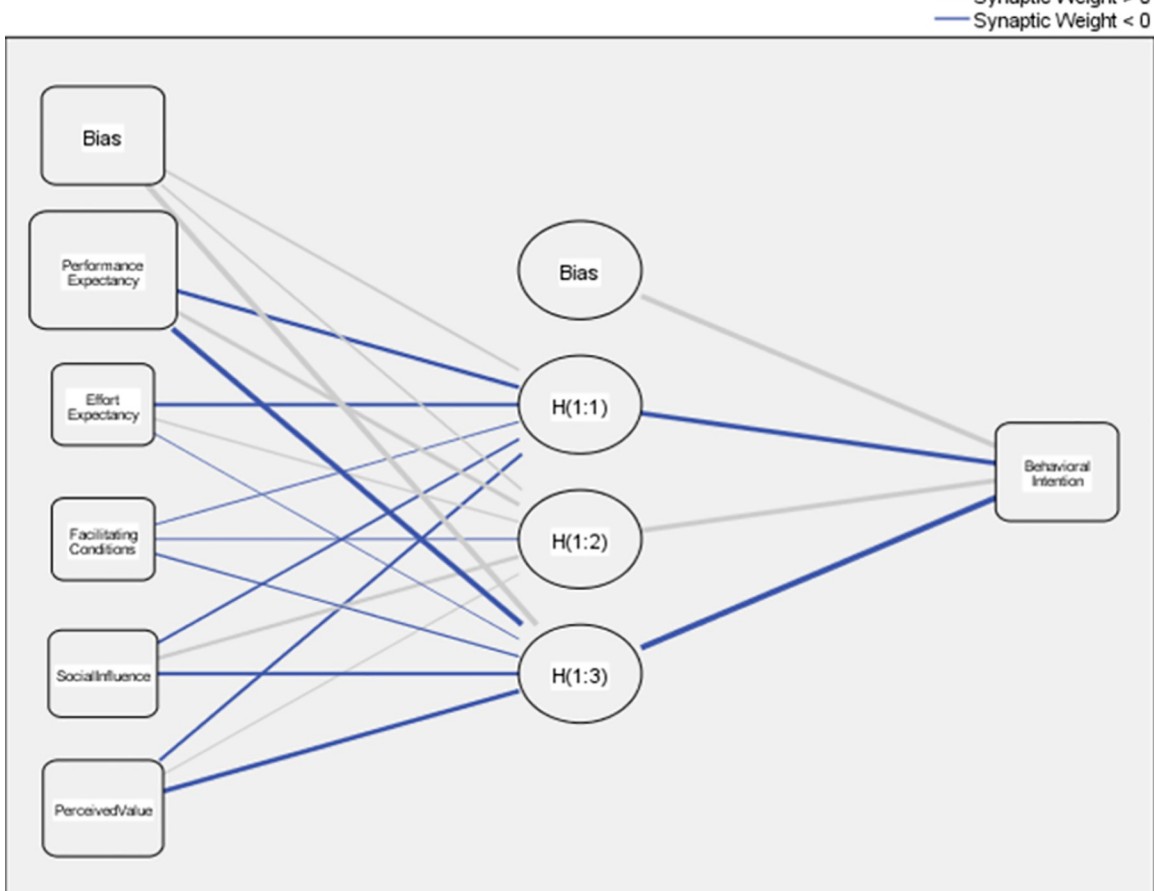

Hidden layer activation function: Sigmoid

Output layer activation function: Sigmoid

**Fig 3. Model A.**

### Results of SEM-hypothesis and ANN tests

As seen in Table 3, the constructs having positive and significant relationships with BI construct and supported hypotheses are as follows: PE ($\beta$ = 0.463, t = 10.68, p = 0.000) $H_1$, EE ($\beta$ = 0.0098, t = 2.52, p = 0.012) $H_2$, FC ($\beta$ = 0.103, t = 2.85, p = 0.004) $H_3$, SI ($\beta$ = 0.118, t = 3.05, p = 0.002) $H_4$, PV ($\beta$ = 0.242, t = 5.39, p = 0.000) $H_6$. There was no significant relationship found between A and BI ($\beta$ = 0.013, t = 0.3, p = 0.764) and $H_5$ was not supported.

As seen in Table 5, according to the results of ANN test, importance order of the independent constructs those are important for BI (Fig 3 Model A) is like the following: PE ($\beta$ = 0,52, NI = 100%), PV ($\beta$ = 0,24, NI = 46,5%), SI ($\beta$ = 0,10, NI = 20,6%), FC ($\beta$ = 0,07, NI = 13,5%), and EE ($\beta$ = 0,06, NI = 12,7%). The most important construct is PE and second one is PV. These results coincided with the SEM results.

As seen in Table 3, the constructs having positive and significant relationships with UB factor and supported hypotheses are as follows: BI ($\beta$ = 0.462, t = 8.87, p = 0.000) $H_7$, U ($\beta$ = 0.188, t = 7.65, p = 0.000) $H_8$, HM ($\beta$ = 0.277, t = 4.90, p = 0.000) $H_9$.

As seen in Table 5, according to the results of ANN test, the importance order of independent construct for UB (Fig 4 Model B) is like the following: BI ($\beta$ = 0,47, NI = 100%), HM ($\beta$ =

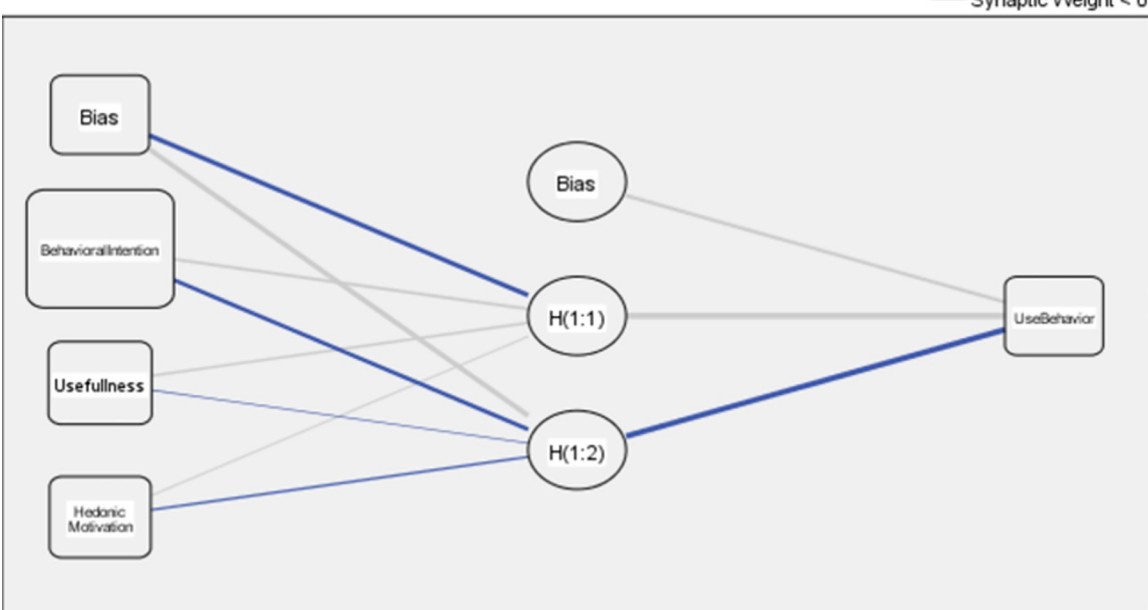

Hidden layer activation function: Sigmoid

Output layer activation function: Sigmoid

**Fig 4. Model B.**

0,31, NI = 67%), and U ($\beta$ = 0,22, NI = 45,5%). The most important independent construct for UB is BI, a second important one is U. The results of Model B partially coincide with the SEM results. According to the SEM results, U was a second important factor, whereas HM was a more important factor according to the ANN test.

In terms of indirect effect, it was determined that HM has a mediator effect in the relationship between U and UB ($\beta$ = 0,20, t = 5.04, p = 0.000), and $H_{10}$ is supported.

**Table 4. ANN analysis average RMSEs and $R^2$.**

| Training | Model A-BI | | | Model B-UB | | |
|---|---|---|---|---|---|---|
| Steps | Training RMSE | Testing RMSE | $R^2$ | Training RMSE | Testing RMSE | $R^2$ |
| ANN1 | 0,0951 | 0,0844 | 0,9910 | 0,0972 | 0,0887 | 0,9906 |
| ANN2 | 0,0926 | 0,0942 | 0,9914 | 0,0978 | 0,0881 | 0,9904 |
| ANN3 | 0,0929 | 0,0964 | 0,9914 | 0,0941 | 0,0983 | 0,9911 |
| ANN4 | 0,0875 | 0,1034 | 0,9923 | 0,0975 | 0,0892 | 0,9905 |
| ANN5 | 0,0943 | 0,0896 | 0,9911 | 0,0906 | 0,1062 | 0,9918 |
| ANN6 | 0,0968 | 0,0859 | 0,9906 | 0,0932 | 0,1002 | 0,9913 |
| ANN7 | 0,0957 | 0,0841 | 0,9908 | 0,1023 | 0,0769 | 0,9895 |
| ANN8 | 0,0914 | 0,0937 | 0,9916 | 0,0967 | 0,1030 | 0,9906 |
| ANN9 | 0,0908 | 0,0957 | 0,9917 | 0,1007 | 0,0817 | 0,9899 |
| ANN10 | 0,1061 | 0,1095 | 0,9887 | 0,0988 | 0,0904 | 0,9902 |
| Average RMSE | 0,094 | 0,094 | 0,991 | 0,097 | 0,092 | 0,991 |
| Standard Deviation | 0,005 | 0,008 | 0,001 | 0,003 | 0,009 | 0,001 |

**Table 5. The average loadings and normalized importance of constructs.**

| Training | Model A-BI | | | | | Model B-UB | | |
|---|---|---|---|---|---|---|---|---|
| Steps | PE | EE | FC | SI | PV | BI | U | HM |
| ANN1 | 0,496 | 0,071 | 0,096 | 0,107 | 0,229 | 0,495 | 0,199 | 0,305 |
| ANN2 | 0,522 | 0,086 | 0,061 | 0,066 | 0,265 | 0,411 | 0,255 | 0,334 |
| ANN3 | 0,500 | 0,041 | 0,059 | 0,170 | 0,231 | 0,480 | 0,265 | 0,255 |
| ANN4 | 0,529 | 0,095 | 0,061 | 0,074 | 0,242 | 0,505 | 0,186 | 0,310 |
| ANN5 | 0,547 | 0,041 | 0,049 | 0,118 | 0,245 | 0,437 | 0,207 | 0,356 |
| ANN6 | 0,480 | 0,066 | 0,083 | 0,112 | 0,259 | 0,492 | 0,218 | 0,290 |
| ANN7 | 0,505 | 0,059 | 0,081 | 0,123 | 0,232 | 0,472 | 0,174 | 0,354 |
| ANN8 | 0,531 | 0,078 | 0,073 | 0,099 | 0,220 | 0,434 | 0,231 | 0,334 |
| ANN9 | 0,556 | 0,054 | 0,061 | 0,085 | 0,245 | 0,516 | 0,212 | 0,272 |
| ANN10 | 0,555 | 0,053 | 0,069 | 0,092 | 0,231 | 0,486 | 0,183 | 0,331 |
| Average Loadings | 0,522 | 0,064 | 0,069 | 0,105 | 0,240 | 0,473 | 0,213 | 0,314 |
| Normalized Importance of Constructs | 100% | 12,7% | 13,5% | 20,6% | 46,5% | 100,0% | 45,5% | 67,1% |

Note. PE: performance expectancy; EE: effort expectancy; FC: facilitating conditions; SI: social influence; PV: perceived value; BI: behavioral intention; U: usefulness; HM: hedonic motivation.

## Discussion and conclusion

### Discussion

This study aims to determine the behavioral intention and the use behavior of the nursing students towards AR technology and the factors affecting them. The SEM and ANN modeling utilized in this study aids in understanding the aspects that drive academic use of AR technology. According to the SEM and ANN findings, as seen in Tables 3 and 5, the most important factor affecting the behavioral intention of students is PE. The students expect that the use of AR will increase their learning/academic performance. Other studies also showed that AR technology is suitable for self-learning methods and increases academic performance [10, 38, 39].

As seen in the results of SEM and ANN in Tables 3 and 5, the second important factor is PV. The students think that AR applications are worth the money when the added value of the benefits is considered. It is consistent with previous research findings [40, 50, 51]. The next important factor in BI is SI. It is observed that SI is more effective when the technology is new and disseminating, but when it becomes public the effect of SI decreases [34]. Considering that the students must keep the social distance due to the Covid-19 pandemic [78], it is estimated that an environment where the effect of SI can be fully activated does not occur. It is predicted that SI may be a more efficient construct in future studies about the acceptance of AR technology after the Covid-19 pandemic.

It was also observed that FC does not have a powerful effect on BI. The main reason for this might be that AR applications used in the study are very user-friendly. Another result supporting this finding is that AR anticipated to be caused by the new technology was not found to have a statistically significant effect. It can be argued that the students are familiar with mobile technologies and this technology does not cause them any anxiety.

The research results show that excessive use of mobile technology in teenagers may cause specific problems such as anti-sociability [79], technology addiction [80] whereas the students find using mobile technology enjoyable [81]. Some studies show that mobile technology addiction negatively affects the academic performances of students [82]. The results of this study differ from previous research results [45–48]. Even though using AR technology creates anxiety;

both the expected benefit from its use and its acceptance are high. Also, using AR applications motivate students.

As seen in Tables 3 and 5, the results show that the most significant factor affecting UB is BI. The students' intentions to use AR technology are strong for both the present and the future. The second important factor affecting UB is HM. The students enjoy using AR technology. The main reason for this is that AR technology allows students to play an active role in the application. The research results overlap with the results of other studies related to AR technology [29, 30] and the enjoyability of mobile technology [81]. As a brand-new mobile technology, AR motivates students with many practical features.

The U feature of AR affecting both UB and HM is an important factor. The fact that it presents the information needed for the content of AR applications and provides convenience and flexibility in accessing information both motivates the use of AR and strengthens the user behavior.

Furthermore, as seen in Tables 3 and 4, the $R^2$ of the ANN models are much greater than the $R^2$ of the PLS-SEM study. This suggests that the variations of BI, and UB to utilize in this study are better explained by the ANN architectures. We believe that the higher $R^2$ values obtained from ANN research are connected to the ANN architecture's capability for deep learning and capturing non-linear correlations between components. Researchers are thus encouraged to carefully watch and address the non-linearity issue using a multi-stage data analysis with deep learning.

## Conclusion

The environment of higher education has evolved dramatically, and internet technologies have played a critical role. With the advent of mobile technologies, students and teachers may now produce their own material and interact with other users; when utilized appropriately, these elements have enormous potential to improve the self-learning experience. Despite the promise for self-learning tools, such as AR technology, to assist the self-learning process, their use in virtual classrooms has not made major gains. It is predicted that the self-learning approach created by the Covid-19 pandemic will maintain its domination in the education system worldwide. For example, in Turkey, the Council of Higher Education has recommended to all universities that 40% of the courses opened after the pandemic should be carried out online. Also, it has paved the way for students to take online courses from national and international universities every semester [12]. The purpose here is to create a flexible education and training system against crises-like situations. With this change, certainly, conventional tools of classroom-based education will not be sufficient and new tools will become widespread. AR applications are mostly developed for self-learning. They can also be developed to improve distant online learning methods. Besides enabling students to practice individually, AR technology also has the potential to create virtual classrooms that multiple users can attend and interact simultaneously. Allowing different users to work and practice on a shared case would certainly differentiate AR technology from other conventional online learning applications. Such a feature would provide convenience not only for clinical nursing practices but also for all health education fields.

Both the changes in the educational policies of the states and the improvements in technology are indicators that the market share of AR and similar products in the education sector will increase. The AR technology market, which was 20 billion USD in 2020, is predicted to be 73 billion in the year 2024 [83]. When the growth potential of the market is taken into consideration, the studies on user expectation like our study are considered of great importance for the developers and manufacturers of AR technology.

## Implications for nursing education

The nursing students think that AR technology will contribute to their academic performances. Although this study is only based on nursing students, the results give ideas about the other areas of health education too. The fact that AR technology is ready to be used anywhere with a mobile phone provides both accessibility and flexibility, and its feature of actively involving the user motivates students. It is predicted that the demand for AR applications will be high because it presents a holistic way of teaching for certain courses and creates an alternative to conventional textbooks.

AR applications that can be used not only by students but also by teachers allowing interactive online education can be developed. The students can improve their clinical nursing skills in addition to learning theoretical knowledge and techniques using AR technology. Not only may the adoption of AR technology assist students or trainees in learning effectively under limited clinical nursing practices, but it can also assist healthcare workers in conducting training and professional skill reinforcement.

We would also like to point out that although the results of this research indicate strong acceptance, teachers must be cautious about the idea that nursing students can learn clinical nursing practices alone, and online with AR without any guidance. It must be kept in mind that strong guidance is essential for nursing education. If nursing education teachers wish to motivate their students and provide them with the best possible learning experiences, AR technology can be utilized, and AR applications can be used as an alternative to textbooks within the next few years.

## Limitations

This study has certain limitations. First, it was based on the experiences of students using AR technology voluntarily for the study. Since it was not mandatory and the students were not held responsible for using the AR technology, how much they use it, how and how much it has affected their academic performance could not be tested. Instead, the general acceptance status of nursing students who have experienced AR technology in their educational life has been examined.

The participants self-reported their intention to use and use behavior towards AR technology. The effect of using AR technology on academic performance was not measured and compared by separating participants into experimental and control groups in any way.

The third limitation is that the AR technology was only used with the mobile phone. No AR laboratories or other AR devices such as headsets, smart glasses were available.

## Supporting information

**S1 Data.**
(CSV)

## Author Contributions

**Conceptualization:** Pelin Uymaz, Ali Osman Uymaz.

**Data curation:** Pelin Uymaz.

**Formal analysis:** Ali Osman Uymaz.

**Methodology:** Pelin Uymaz, Ali Osman Uymaz.

**Resources:** Ali Osman Uymaz.

**Software:** Ali Osman Uymaz.

**Supervision:** Pelin Uymaz.

**Validation:** Ali Osman Uymaz.

**Visualization:** Pelin Uymaz, Ali Osman Uymaz.

**Writing – original draft:** Pelin Uymaz, Ali Osman Uymaz.

**Writing – review & editing:** Pelin Uymaz, Ali Osman Uymaz.

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
