## [Decision Letter · Decision Letter 0]

17 Nov 2021

PONE-D-21-32878ASSESSING ACCEPTANCE OF AUGMENTED REALITY IN NURSING EDUCATIONPLOS ONE

Dear Dr. UYMAZ,

Thank you for submitting your manuscript to PLOS ONE. After careful consideration, we feel that it has merit but does not fully meet PLOS ONE’s publication criteria as it currently stands. Therefore, we invite you to submit a revised version of the manuscript that addresses the points raised during the review process.

We look forward to receiving your revised manuscript.

Kind regards,

Gwo-Jen Hwang

Academic Editor

PLOS ONE

Journal Requirements:

Reviewers' comments:

Reviewer's Responses to Questions

**Comments to the Author**

1. Is the manuscript technically sound, and do the data support the conclusions?

Reviewer #1: Yes

Reviewer #2: No

2. Has the statistical analysis been performed appropriately and rigorously? 

Reviewer #1: Yes

Reviewer #2: No

3. Have the authors made all data underlying the findings in their manuscript fully available?

Reviewer #1: Yes

Reviewer #2: No

4. Is the manuscript presented in an intelligible fashion and written in standard English?

Reviewer #1: Yes

Reviewer #2: Yes

5. Review Comments to the Author

Reviewer #1: The authors aimed to investigate the acceptance of AR in nursing education. They used s a quantitative research method by collecting 419 nursing students’ online feedback to the questionnaire items. In addition to examining the linear relationships between using the structural equation modeling, they also used artificial neural networks to determine the non-linear relationships between constructs.

The topic is interesting and the study is generally well conducted. However, the background and literature review sections are weak. The authors need to state in detail why investigating nursing students’ AR acceptance and the use of ANN are needed by referring to the literature. Some recommended references are listed as follows:

Yuksekdag, B. B. (2018). The Importance of Mobile Augmented Reality in Online Nursing Education. In Nursing Education, Administration, and Informatics: Breakthroughs in Research and Practice (pp. 111-125). Igi Global.

Chang, C. Y., Lai, C. L., & Hwang, G. J. (2018). Trends and research issues of mobile learning studies in nursing education: A review of academic publications from 1971 to 2016. Computers & Education, 116, 28-48.

Hwang, G. J., Xie, H., Wah, B. W., & Gašević, D. (2020). Vision, challenges, roles and research issues of artificial intelligence in education. Computers & Education: Artificial Intelligence, 1, 100001.

Chen, X., Xie, H., Zou, D., & Hwang, G. J. (2020). Application and theory gaps during the rise of Artificial Intelligence in Education. Computers and Education: Artificial Intelligence, 1, 100002.

Reviewer #2: DEAR Author,

The manuscript “ASSESSING ACCEPTANCE OF AUGMENTED REALITY IN NURSING EDUCATION’ is quite interesting, but for contribution some sections should clarify for publication.

- Why focus on human anatomy with AR in nursing education, it’s unclear? Suggest auditors add references to support the issue with human anatomy in nursing education.

-Page 4, the 2. Theory section, suggest a change subject title to Background or Literature review, it’s would be better.

-Section 3.3. Sample and Data Only presented “The Research population for this study was nursing students. The data were collected online from 446 nursing students by using simple random sampling between November 2020 and January 2021.” How to conduct AR with nursing class with intervention process is very vague.

--Page13~16, the context contribution related to all ANN described, it’s not clear for the reader, and include tables, for example, what’s the ANN1,2,3…….10? if authors want to keep the section, please address more on the context related to all ANN.

--The context data of “Table 3. The Research Model Evaluation Results” do not match Figure 2. The Research Model Path Analysis, please carefully clarify. So the section needs rewording based on the revised data.

- Figure 3. Model A and Figure 4. Model B, they are so vague, please address more.

-Based on the incorrect data, please reword Discussion and Conclusion section.

---

## [Author Response · Author response to Decision Letter 0]

6 Dec 2021

Dear Editor,

We appreciate you and the reviewers for taking the time to read our work and provide important feedback. Your important and informative feedback has resulted in possible changes in the current version. We attentively considered the comments and did our best to address each one. We hope that the manuscript, following rigorous edits, meets your high expectations. If there are any further constructive remarks, the writers would appreciate them. The point-by-point replies were provided in the file of response to reviewers.

---

## [Decision Letter · Decision Letter 1]

10 Jan 2022

PONE-D-21-32878R1ASSESSING ACCEPTANCE OF AUGMENTED REALITY IN NURSING EDUCATIONPLOS ONE

Dear Dr. UYMAZ,

Thank you for submitting your manuscript to PLOS ONE. After careful consideration, we feel that it has merit but does not fully meet PLOS ONE’s publication criteria as it currently stands. Therefore, we invite you to submit a revised version of the manuscript that addresses the points raised during the review process.

We look forward to receiving your revised manuscript.

Kind regards,

Heng Luo, Ph.D.

Academic Editor

PLOS ONE

Journal Requirements:

Reviewers' comments:

Reviewer's Responses to Questions

**Comments to the Author**

1. If the authors have adequately addressed your comments raised in a previous round of review and you feel that this manuscript is now acceptable for publication, you may indicate that here to bypass the “Comments to the Author” section, enter your conflict of interest statement in the “Confidential to Editor” section, and submit your "Accept" recommendation.

Reviewer #2: (No Response)

Reviewer #3: (No Response)

2. Is the manuscript technically sound, and do the data support the conclusions?

Reviewer #2: No

Reviewer #3: Yes

3. Has the statistical analysis been performed appropriately and rigorously? 

Reviewer #2: Yes

Reviewer #3: Yes

4. Have the authors made all data underlying the findings in their manuscript fully available?

Reviewer #2: No

Reviewer #3: Yes

5. Is the manuscript presented in an intelligible fashion and written in standard English?

Reviewer #2: Yes

Reviewer #3: Yes

6. Review Comments to the Author

Reviewer #2: Dear author,

- the context contribution is related to all ANN described. Please address more on the context related to the Discussion and Conclusion section.

--The context data of “Table 3. The Research Model Evaluation Results” do not match Figure 2. The Research Model Path Analysis, some data incorrect, please carefully clarify.

- Figure 3. Model A and Figure 4. Model B they are so vague; please address more.

-The authors used artificial neural networks to determine the non-linear relationships between constructs. Please address more in the Discussion and Conclusion section.

Thank you.

Reviewer #3: This paper investigated the acceptance of AR in nursing education. The author collecting 419 nursing students’ feedback through online questionnaire, and then use structural equation modeling and artificial neural networks to examining the linear relationships and non-linear relationships between constructs, respectively. The results were very interesting and contribute to the AR technology application in nurse education.

However, there were several information need to compliment:

First, the part of Introduction.

(1) in the second paragraph of Introduction, mentioned that online learning can provide learning resources, and then emphasized “The nursing students were advised to download Augmented Reality (AR) applications and use them as an auxiliary source while studying clinical nursing practices”, the logic was not clear. Compared to traditional online learning resources, what are the special aspects of augmented reality learning resources that are not discussed clearly. (78-81 line)

(2) in the fourth paragraph of introduction, the necessity of this research is not clearly discussed. What is the difference between the characteristics of AR learning resources and traditional online learning materials? Why does nursing education need AR mobile learning resources, and what defects are made up?

Second, the part of Literature review.

In the literature review of AR technology, mentioned the affordance and strength of the AR technology. However, the relationship between these advantages of AR technology and nursing education is not highlighted, that is, whether there are some difficulties in the process of teaching knowledge in nursing education, and AR technology can provide solutions to these difficulties, and explain the necessity of applying AR technology to nursing education.

Third, the part of Methods.

The methods part introduces the situation of participants and the questionnaire used, but it lacks how the Internal Medicine Nursing Course was carried out during the epidemic, how the learning materials supported by augmented reality technology were designed, and how students used augmented reality materials to carry out learning. It is suggested to add.

Lastly, the part of the conclusion.

The conclusion and introduction are repeated. For example, lines 488-490 and 87-88 of the introduction repeat "The Council of Higher Education has recommended to all universities that 40% of the courses Opened after the pandemic should be carried out online ". But this example doesn't seem to be directly related to augmented reality and the need for AR in nursing education.

7. PLOS authors have the option to publish the peer review history of their article (what does this mean?). If published, this will include your full peer review and any attached files.

Reviewer #2: No

Reviewer #3: No

---

## [Author Response · Author response to Decision Letter 1]

22 Jan 2022

Dear reviewer,

We appreciate you for taking your time to read our work and provide important feedback. Your important and informative feedback has resulted in possible changes in the current version. We attentively considered your comments and did our best to address each one. We hope that the manuscript, following rigorous edits, meets your high expectations. If there are any further constructive remarks, the writers would appreciate them. 

The point-by-point replies are provided below. All changes have been indicated in green in the paper.

Best regards

---

## [Editor Report · Decision Letter 2]

31 Jan 2022

ASSESSING ACCEPTANCE OF AUGMENTED REALITY IN NURSING EDUCATION

PONE-D-21-32878R2

Dear Dr. UYMAZ,

We’re pleased to inform you that your manuscript has been judged scientifically suitable for publication and will be formally accepted for publication once it meets all outstanding technical requirements.

Kind regards,

Heng Luo, Ph.D.

Academic Editor

PLOS ONE
---

## [Editor Report · Acceptance letter]

7 Feb 2022

PONE-D-21-32878R2 

Assessing acceptance of augmented reality in nursing education 

Dear Dr. UYMAZ:

I'm pleased to inform you that your manuscript has been deemed suitable for publication in PLOS ONE. Congratulations! Your manuscript is now with our production department. 

Kind regards, 

on behalf of

Dr. Heng Luo 

Academic Editor

PLOS ONE